# Identification of potential biomarkers in active Lyme borreliosis

Ivar Tjernberg[1,2]*, Malin Lager[3], Guro Furset Jensen[4,5], Randi Eikeland[6,7,8], Dag Nyman[8,9], Lars Brudin[10,11], Anna J. Henningsson[8,12,13]

1 Department of Clinical Chemistry and Transfusion Medicine, Region Kalmar County, Kalmar, Sweden, 2 Department of Biomedical and Clinical Sciences, Division of Inflammation and Infection, Linköping University, Linköping, Sweden, 3 National Reference Laboratory for Borrelia, Department of Clinical Microbiology in Jönköping, Region Jönköping County, Linköping University, Linköping, Sweden, 4 National Reference Laboratory for Borrelia, Department of Clinical Microbiology, Sørlandet Hospital Trust, Kristiansand, Norway, 5 Department of Clinical Microbiology, Sørlandet Hospital Health Enterprise, Kristiansand, Norway, 6 National Advisory Unit on Tick Borne Diseases, Sørlandet Hospital Trust, Kristiansand, Norway, 7 Faculty of Health and Sports Science, University of Agder, Grimstad, Norway, 8 ESCMID Study Group for Lyme Borreliosis—ESGBOR, Part of the European Society for Clinical Microbiology and Infectious Diseases, Basel, Switzerland, 9 The Åland Group for Borrelia Research, Mariehamn, Finland, 10 Department of Clinical Physiology, Region Kalmar County, Kalmar, Sweden, 11 Department of Health, Medicine and Caring Sciences, Linköping University, Linköping, Sweden, 12 National Reference Laboratory for Borrelia and Other Tick-Borne Bacteria, Department of Clinical Microbiology in Jönköping, Region Jönköping County, Linköping University, Linköping, Sweden, 13 Department of Biomedical and Clinical Sciences, Linköping University, Linköping, Sweden

* ivar.tjernberg@liu.se

## Abstract

### Objectives

Lyme serology does not readily discriminate an active Lyme borreliosis (LB) from a previous *Borrelia* infection or exposure. Here, we aimed to investigate a large number of immunological protein biomarkers to search for an immunological pattern typical for active LB, in contrast to patterns found in healthy blood donors, a proportion of whom were previously exposed to *Borrelia*.

### Methods

Serum samples from well-characterised adult patients with ongoing LB and healthy blood donors were included and investigated using a proximity extension assay (provided by Olink®) by which 92 different immune response-related human protein biomarkers were analysed simultaneously.

### Results

In total, 52 LB patients and 75 healthy blood donors were included. The blood donors represented both previously *Borrelia* exposed (n = 34) and not exposed (n = 41) based on anti-*Borrelia* antibody status. Ten of the examined 92 proteins differed between patients and blood donors and were chosen for further logistic regression (p<0.1). Six proteins were statistically significantly different between LB patients and blood donors (p<0.05). These six

**Data Availability Statement:** There are restrictions in place that prevent public sharing of original and raw data according to the regional ethical permissions for the study. Data contain potentially identifying and sensitive patient information and

public sharing is not allowed according to the consents signed by the study participants. For sharing of restricted data, please contact the corresponding author Ivar Tjernberg at ivar.tjernberg@liu.se or Linköping University at registrator@liu.se.

**Funding:** This study was supported by university grants to Region Kalmar County (IT), Futurum-Academy for Healthcare and Division of Medical Diagnostics, Region Jönköping County, Interreg IVA Program ScandTick (grant no. 167226) and Interreg V program ScandTick Innovation (project ID. 20200422, reference no. 2015-29 000167), The Medical Research Council of Southeast Sweden (FORSS-475511), the foundation for medical research of the Åland cultural foundation and the Wilhelm and Else Stockmann Foundation (AJH). The funders had no role in study design, data collection and analysis, decision to publish, or preparation of the manuscript.

**Competing interests:** I have read the journal's policy and the authors of this manuscript have the following competing interests: IT reports participation in advisory board and personal fees from Pfizer Inc outside the submitted work. AJH has a collaborative research agreement with Abbott Laboratories, Chicago, USA and Reagena Oy, Toivala, Finland. Remaining authors report no conflict of interest.

proteins were then combined in an index and analysed using receiver-operating-characteristic curve analysis showing an area under the curve of 0.964 (p<0.001).

## Conclusions

The results from this study suggest that there is an immunological protein pattern that can distinguish a present *Borrelia* infection from a previous exposure as well as anti-*Borrelia* antibody negative blood donors. Although this method is not adapted for routine clinical use at this point, the possibility is interesting and may open new diagnostic opportunities improving the laboratory diagnostics of LB.

## Introduction

Lyme borreliosis (LB) is considered the most common tick-borne disease in both Europe and North America [1–3]. The disease, caused by bacteria belonging to the *Borrelia burgdorferi* sensu lato (s.l.) complex, may give rise to a number of different clinical manifestations including erythema migrans, Lyme neuroborreliosis, acrodermatitis chronica atrophicans, Lyme arthritis, and other more rare manifestations [2]. However, *Borrelia* exposure may also pass sub-clinically in a significant proportion of exposed individuals [3–5]. In the routine clinical setting, diagnosis of erythema migrans relies on patient history and physical examination, whereas in the remaining manifestations laboratory support is also required in the diagnostic workup. Although much effort has been invested in the development of direct detection by molecular methods, these are not widely used due to low sensitivity in some materials, e.g. cerebrospinal fluid. In clinical practice, indirect methods, essentially antibody detection by serology, remains the gold standard for the laboratory diagnosis of LB [2, 6]. However, there are challenges in the interpretation of anti-*Borrelia* antibody results that must be considered in the diagnosis of LB. The serological results must be interpreted in relation to the variable clinical presentation of LB including the dissemination of the infection. It is important to recognise that a single positive serological result does not discriminate an active infection from a previous symptomatic or subclinical exposure [6, 7]. Furthermore, the local epidemiological situation needs to be considered. In highly endemic areas, the seroprevalence may reach high figures in the healthy population thus significantly reducing the positive predictive value of a positive test result [8, 9]. In addition, test cross-reactions may also render false positive serological results [6, 10, 11]. However, the most important limitation of *Borrelia* serology in the routine clinical setting in endemic areas is that the population is highly seropositive after previous exposure to *Borrelia*. Additional tests to discriminate an active LB from a previous *Borrelia* infection would improve the diagnostic process considerably and limit the risk for overdiagnosing and thereby counteract overuse of antibiotics with associated risks for side-effects for patients as well as the negative effects on the environment including antibiotic resistance.

Using proximity extension assay, large numbers of proteins may be detected and measured simultaneously in the same patient sample. Predefined analytical panels of different sizes and combinations of biomarkers are available through Olink®, e.g. neurology, cardiovascular and inflammation panels. For the purpose of this study, the immune response panel was chosen based on the hypothesis that the immune response shows different signatures of biomarkers during an active LB as opposed to a previous *Borrelia* infection.

Thus, in this study, we wanted to investigate serum samples from clinically well characterised LB patients and healthy blood donors examining a large number of immunological protein biomarkers in order to possibly distinguish active LB from previous *Borrelia* exposure.

## Material and methods

### Lyme borreliosis patients and blood donors

Serum samples from a previous prospective study performed 2013–2017 of adult LB patients with ongoing infection and blood donors were included in the present study. The original study included patients from Jönköping County in Sweden, Kristiansand in Norway and on the Åland Islands, and blood donors from Jönköping County, Sweden [12]. In the original study 59 patients with various manifestations of LB were included, of whom seven patients were considered possible Lyme neuroborreliosis. Only the 52 LB patients with a definite diagnosis were included in this study. The 52 samples represented patients with different ongoing manifestations of LB. Serum samples were obtained from the LB patients at the time of diagnostic investigation, thus with ongoing symptoms/complaints, but before treatment initiation, thereby considered active/ongoing LB. Classification and confirmation of LB diagnosis in the patients were based on clinical information including laboratory results and review of medical records by experienced physicians in the field. Patients with Lyme neuroborreliosis fulfilled the European case definition for definite Lyme neuroborreliosis [13]. Patients with Lyme arthritis and/or acrodermatitis chronica atrophicans showed clinical signs compatible with the conditions and were all confirmed with detection of *Borrelia*-specific DNA. Patients with erythema migrans reported recent tick-bite and presented with a typical skin rash of >5 cm in diameter assessed by a physician [12].

Furthermore, a random subset (n = 80) of the original 201 blood donor samples were included as a healthy comparison group, thus considered not having ongoing LB. Blood donors were all healthy at the time of sampling and only included after an approved health declaration without current symptoms or signs of disease. The health declaration includes questions regarding potential fever, infection, other diseases, medical examinations, treatments, medications. Information regarding possible previous LB was gathered from the blood donors. Both blood donors with and without anti-*Borrelia* antibodies were included in order to also cover individuals with a previous exposure to *B. burgdorferi* s.l. The blood donors were screened for anti-*Borrelia* antibodies using Enzygnost Borrelia Lyme IgM/IgG (Siemens/DADE Behring, Marburg, Germany); for IgG based on a mix of native *Borrelia* antigens from *B. afzelii* strain PKo and recombinant VlsE obtained from genospecies the *B. burgdorferi* sensu stricto, *B. garinii* and *B. afzelii*. These 80 blood donors comprised 37 IgM and/or IgG anti-*Borrelia* antibody positive (34 IgG-positive and three IgM and IgG-positive) and 43 anti-*Borrelia* antibody negative subjects. The IgG antibody reactivities in these 37 samples were also confirmed by positive findings in at least one additional and independent *Borrelia* serological assay including Liaison Borrelia IgG (DiaSorin, IgG-analysis based on recombinant *Borrelia*-specific VlsE antigens from *B. garinii* strain PBi) and the Immunetics C6 LYME ELISA (IgM/IgG analysis based on a synthetic 25 amino acid sequence derived from IR6 of VlsE *B. burgdorferi* strain B31). All dilution steps in the antibody assays were performed according to the respective manufacturer's instructions. After classification of LB patients and blood donors data was anonymised. Further details and definitions of the study subjects have been described in a previous publication [12].

### Laboratory analyses

Serum samples were all investigated using a proximity extension assay provided by the Clinical Biomarkers facility, Science for Life Laboratory, Uppsala University, Sweden. Detailed

description on the principle of the method has been published previously [14]. In short, the 92 human immune system biomarkers in the chosen panel are detected in parallel by individually paired matched antibodies labelled with unique strands of DNA oligonucleotides that may hybridize when in proximity with each other, allowing for subsequent amplification, e.g. using quantitative PCR, in which the results are related to the initial concentration of each target. The panel used was the immune response panel v.3202 covering 92 immune response-related human protein biomarkers. A comprehensive list of analytes covered in the panel may be found at the Olink website [15]. Proteins in this publication have been referenced by their respective gene as well as their unique protein identification number (UniProt ID) according to the Universal Protein Resource (www.uniprot.org).

## Data analysis and statistics

Results for all parameters were presented as normalised protein expression values on a logarithmic scale with base 2 ($\log_2$) based on the respective analytical cycle threshold values.

The lowest concentration possible to detect (LOD) for each of the proteins is given. Individual values below LOD are replaced by LOD-1 which represents 50% of LOD. The number of values below LOD (%) per protein are presented in the S1 Table. Parameters with at least 50% values within the detectable range were chosen for the subsequent analysis. The further analyses were as follows: 1) The database was randomly divided in two halves, equally for both LB patients and blood donors, and the difference between the two groups within each half analysed separately with Mann-Whitney's U-test. This was done in order to reduce random statistical effects. 2) Secondly, the differences between LB patients and blood donors, with a significance level of <0.1 simultaneously for the two groups were qualified for successive analyses with multivariate logistic regression with p<0.05 as a criterion. These were then subjected to form an index using a linear equation with the log(OR) as weights for the various proteins, then presented in receiver-operating characteristic (ROC) analysis. Furthermore, the possibility of an age and/or sex impact on the index was investigated on the odds ratios. Statistical analyses were performed in Statistica version 13.

## Ethics

The study was approved by the respective regional ethical review boards in Sweden, Åland and Norway. For Sweden 2013/238-31, 2014/326-32, 2016/211-31, 2019–03832, for Åland the approval was given at the regional ethical review board meeting no 2/2014 and for Norway 2014/1100 and additionally September 13, 2019 with reference no 9463. All study participants agreed to take part in the study and signed a written informed consent form.

## Results

A complete set of laboratory results with approved quality control results were obtained from all 52 LB patients and 75 of the original 80 blood donors. Thus, five blood donors with incomplete or failed results were excluded from the study (three anti-*Borrelia* antibody positive and two anti-*Borrelia* antibody negative). Basic demographics of the LB patients and the remaining blood donors are shown in Table 1.

The LB patients (n = 52) consisted of the following manifestations; Lyme neuroborreliosis (n = 41), acrodermatitis chronica atrophicans and/or Lyme arthritis (n = 7) and erythema migrans (n = 4). Further details regarding the subjects may be found in the publication by Lager *et al.* [12]. Thus, LB patients were all sampled during an ongoing manifestation of LB, while the blood donors were all healthy at sampling. Median normalised protein expression values in $\log_2$ scale together with quartile one-three results for the included 92 parameters

**Table 1. Demographic characteristics of Lyme borreliosis patients (n = 52) and blood donors (n = 75).**

| Clinical group | Lyme borreliosis patients (n = 52) | | | Blood donors (n = 75) | |
|---|---|---|---|---|---|
| | Definite LNB (n = 41) | ACA/LA (n = 7) | Erythema migrans (n = 4) | Blood donors AB+ (n = 34) | Blood donors AB− (n = 41) |
| Sex, female/male (% female) | 25/16 (70) | 4/3 (57) | 3/1 (75) | 6/28 (18) | 13/28 (32) |
| Median age in years (range) | 61 (21–85) | 48 (27–55) | 51 (45–69) | 44 (20–68) | 48 (21–66) |
| Self-reported previous LB (%) | N/A | N/A | N/A | 6 (18) | 2 (5) |

LNB, Lyme neuroborreliosis; ACA, Acrodermatitis chronica atrophicans; LA, Lyme arthritis; AB+, anti-*Borrelia* antibody positive; AB−, anti-*Borrelia* antibody negative;
LB, Lyme borreliosis

N/A, not available

from the remaining 52 LB patients and 75 blood donors (34 anti-*Borrelia* antibody positive and 41 anti-*Borrelia* antibody negative) are shown in S1 Table. In total, ten of the 92 variables showed a difference with p<0.1, further included in a logistic regression model in which six (PPP1R9B, PRDX5, ITM2A, EIF4GI, DDX58, ITGB6) of the ten protein markers remained (p<0.05), Table 2. Furthermore, an index was created using the individual results for these six protein markers:

$$\text{Index} = \text{PPP1R9B}*(-1.962) + \text{PRDX5}*(2.307) + \text{ITM2A}*(-2.632) + \text{EIF4G1}*(-2.477) \\ + \text{DDX58}*(2.295) + \text{ITGB6}*(-2.121)$$

For comparison between LB patients and blood donors, this index was used in a subsequent ROC curve analysis shown in Fig 1. A significant difference (p<0.001) was shown with an area under curve (AUC) of 0.964. In addition, two further ROC curve analyses of subgroups of patients and blood donors were performed as follows: a) LB patients (n = 52) and the subgroup of anti-*Borrelia* antibody positive blood donors (n = 34) showed an AUC of 0.973 and b) Lyme neuroborreliosis patients (n = 41) and all blood donors (n = 75) resulting in an AUC of 0.962. No statistically significant differences were found comparing these three AUCs.

Using an index criterion value of >−8.8178 to discriminate LB patients from blood donors in the original ROC curve analysis, a sensitivity of 86% and a specificity of 92% could be achieved. As shown by the index calculation, high values of PRDX5 and DDX58 in

**Table 2. Median (Q1-Q3) of the ten proteins qualified for the successive multivariate logistic regression.** OR is odds ratio with 95% confidence limits where control is reference.

| Protein | Uniprot_ID | Lyme borreliosis patients n = 52 | Blood donors n = 75 | p-value* | Odds ratio (95% CL) | p-value |
|---|---|---|---|---|---|---|
| CKAP4 | Q07065 | 4.10 (3.96–4.33) | 3.93 (3.72–4.15) | 0.071 | | |
| DDX58 | O95786 | 2.43 (2.00–2.89) | 1.95 (1.75–2.40) | 0.037 | 9.93 (1.49–66.21) | 0.018 |
| EIF4G1 | Q04637 | 2.30 (1.84–2.84) | 2.59 (2.25–3.03) | 0.087 | 0.08 (0.01–0.58) | 0.013 |
| HEXIM1 | O94992 | 3.27 (2.95–3.46) | 3.39 (3.10–3.85) | 0.052 | | |
| IL6 | P05231 | 3.09 (2.51–3.55) | 2.29 (1.87–2.73) | 0.004 | | |
| ITGB6 | P18564 | 2.66 (2.37–2.98) | 3.03 (2.76–3.23) | 0.004 | 0.12 (0.01–0.97) | 0.046 |
| ITM2A | O43736 | 3.06 (2.67–3.45) | 3.75 (3.41–4.03) | 0.012 | 0.07 (0.01–0.41) | 0.003 |
| PPP1R9B | Q96SB3 | 2.12 (1.09–2.62) | 2.59 (2.29–2.98) | 0.076 | 0.14 (0.04–0.51) | 0.003 |
| PRDX5 | P30044 | 5.57 (4.93–6.17) | 4.77 (4.24–5.26) | 0.016 | 10.0 (2.0–50.6) | 0.006 |
| SH2B3 | Q9UQQ2 | 0.98 (0.98–2.22) | 2.33 (0.98–2.93) | 0.008 | | |

*) p-values from Mann-Whitneys U-test (<0.1), necessary for inclusion into the succeeding logistic regression (for details see "Methods").

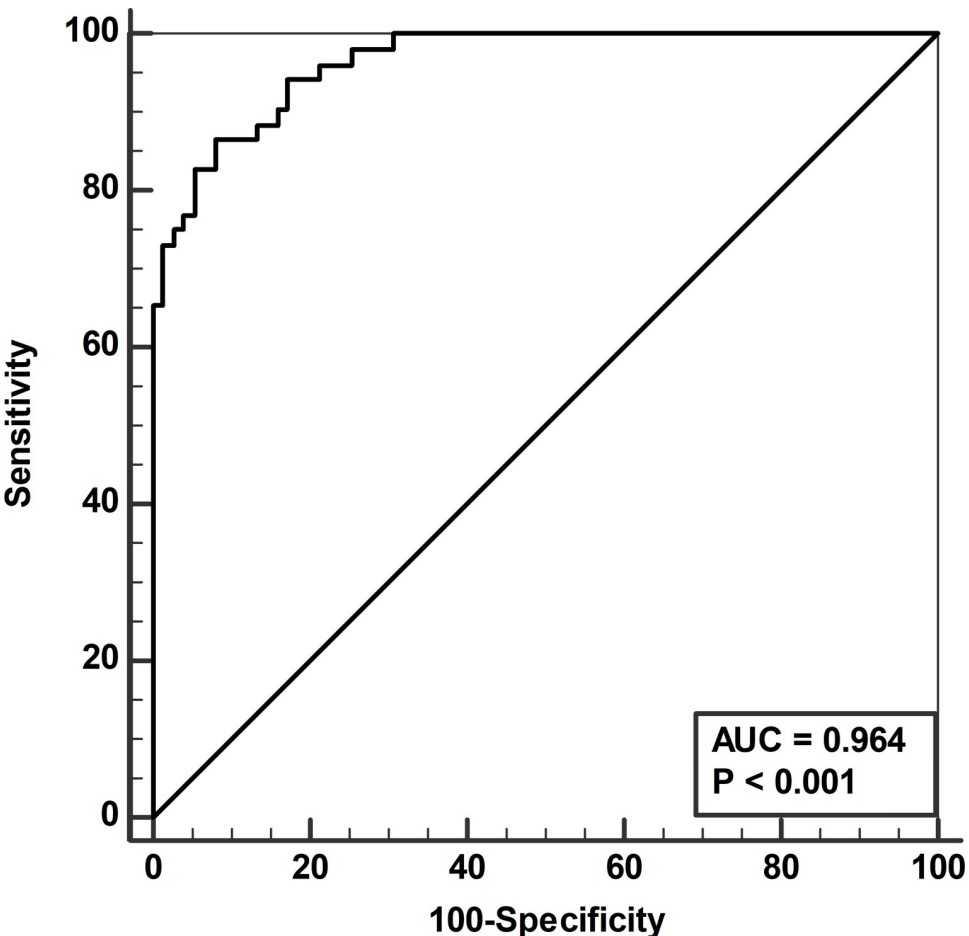

**Fig 1. Receiver operating characteristic curve displaying the diagnostic power in the six protein index discriminating Lyme borreliosis patients from blood donors.** AUC, area under curve.

combination with low values for the remaining markers were typical for the LB patients, while the opposite pattern was characteristic for the blood donors. Age and sex had no impact on the odds ratio of the index, p-values for odds ratios were for age 0.68 and for sex 0.73.

## Discussion

We here show the possibility to successfully distinguish active ongoing LB from blood donors with and without previous exposure to *Borrelia* using a multiplex biochemical approach independent from *Borrelia* serology. The results in this study imply that a number of immune-related serum protein biomarkers may serve this purpose.

In detail, the results show that six proteins are indicated, namely the gene products corresponding to PPP1R9B, PRDX5, ITM2A, EIF4GI, DDX58, and ITGB6. These proteins serve different functions in the human body briefly described here. PPP1R9B, protein phosphatase 1 regulatory subunit 9B, also named neurabin2 and spinophilin seems to be involved in multiple signalling pathways as a scaffold protein. Furthermore, it has been reported to play a significant role as a link between actin cytoskeleton and the plasma membrane at synaptic junctions as well as a G-protein signalling regulator. It also appears that this protein is of special importance in the nervous system [16, 17]. PRDX5, peroxiredoxin-5, also known as PMP20 is a

thiol-specific peroxidase that catalyses reduction of organic hydroperoxides and hydrogen peroxide thereby playing a role in cell protection against oxidative stress [17, 18]. ITM2A, integral membrane protein 2A, has been reported to play a role in regulating the development and function of T cells, but additional data on the function of this protein is lacking [19]. EIF4GI, eukaryotic translation initiation factor 4 gamma 1, constitutes a part of the protein complex eIF4F, central for ribosome protein translation initiation and its function thereby regulates protein synthesis. Several viruses have been reported to modify the function of EIF4G by proteolytic cleavage resulting in down-regulation of cellular translation initiation [20], thus a target protein that may be relevant for host defence in infections. DDX58 corresponds to an antiviral innate immune response receptor retinoic acid-induced gene-I (RIG-I), that senses viral nucleic acids in the cytoplasm and further activates a signalling cascade that leads to the production of pro-inflammatory cytokines and type I interferons, which also may play important roles in bacterial infections [21]. Finally, ITGB6, integrin subunit beta 6, constitutes a part of the heterodimeric integrin αvβ6 transmembrane receptor exclusively expressed in epithelial cells. Integrin αvβ6 may serve as a receptor for fibronectin, but one of its major functions is to activate transforming growth factor-β1, a key regulator of the innate anti-inflammatory surveillance of the body [22]. Taken together, the combined profile suggests that these six proteins may represent relevant reactions caused by the pathogen in combination with the host immune defence thus including cellular signal transduction, regulation of oxidative stress and protein synthesis, and regulation of cellular immunity as well as inflammatory and anti-inflammatory reactions necessary for clearance of the pathogen. Interestingly, the markers may be both up- and downregulated, and in ongoing LB both PRDX5 and DDX58 were upregulated while the remaining markers were downregulated, suggesting importance of this specific pattern in the present ongoing LB infection phase. To our knowledge, these six markers have not previously been investigated and shown relevant in the perspective of LB. Thus, the significance of them in the discrimination of active from previous infection needs further confirmation in other clinical LB materials as well as in differential conditions to LB including other infectious and inflammatory conditions. Furthermore, longitudinal studies of confirmed LB patients with follow-up sampling after treatment and resolution of symptoms would be of great value as well as patients classified after additional and other LB serological assays.

The strengths of this study include the well-characterised and relevant clinical LB patients together with subjects representing both healthy controls (blood donors) with and without anti-*Borrelia* antibodies. Moreover, the included LB patients represent the major and most important of the various clinical manifestations of LB in Europe. Additionally, the proximity extension assay methodology applied to the serum material is sensitive and specific [14]. Regarding limitations, although a reasonable number of LB patients and blood donors were investigated, the material is still limited in number, and follow-up samples were not available for the patients which would have been a valuable source of confirmation. Also, the number of studied parameters in relation to the number of available samples needs to be handled with caution in order to avoid random statistical effects.

To conclude, we here show a possibility to discriminate ongoing LB from previous exposure to *Borrelia* using a powerful multiplex serum protein approach. This new strategy is interesting and opens for improvements in the diagnostics of LB in principle although not adapted for clinical use at this point. Further studies on well-characterised clinical materials including follow-up samples and relevant control conditions are warranted. Additional screening using multiplex technology may confirm our data and possibly also detect new and fewer biomarkers with the same diagnostic power suitable for routine use.

## Supporting information

**S1 Table. <LOD; Frequency of samples with results below the lowest concentration possible to detect.** *Highest p-value below 0.1 of two randomly selected halves of the study population calculated using Mann-Whitney's U-test (for details see the original publication). (DOCX)

## Acknowledgments

The authors would like to acknowledge support of the Clinical biomarker facility at SciLifeLab Sweden for providing assistance in protein analyses. We would also like to thank the patients and blood donors for participating in this study as well as all involved health care professionals and departments for valuable assistance.

## Author Contributions

**Conceptualization:** Ivar Tjernberg.

**Data curation:** Ivar Tjernberg, Malin Lager, Guro Furset Jensen, Randi Eikeland, Dag Nyman, Lars Brudin, Anna J. Henningsson.

**Formal analysis:** Ivar Tjernberg, Lars Brudin.

**Funding acquisition:** Ivar Tjernberg, Anna J. Henningsson.

**Investigation:** Malin Lager, Guro Furset Jensen, Randi Eikeland, Lars Brudin, Anna J. Henningsson.

**Methodology:** Ivar Tjernberg, Randi Eikeland, Dag Nyman, Lars Brudin, Anna J. Henningsson.

**Project administration:** Ivar Tjernberg.

**Resources:** Malin Lager, Guro Furset Jensen, Randi Eikeland, Dag Nyman, Anna J. Henningsson.

**Supervision:** Ivar Tjernberg.

**Writing – original draft:** Ivar Tjernberg.

**Writing – review & editing:** Ivar Tjernberg, Malin Lager, Guro Furset Jensen, Randi Eikeland, Dag Nyman, Lars Brudin, Anna J. Henningsson.

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
