## [Decision Letter · Decision Letter 0]

30 Mar 2023

PONE-D-23-05893A novel laboratory approach to discriminate active Lyme borreliosis from previous Borrelia exposurePLOS ONE

Dear Dr. Tjernberg,

Thank you for submitting your manuscript to PLOS ONE. After careful consideration, we feel that it has merit but does not fully meet PLOS ONE’s publication criteria as it currently stands. Therefore, we invite you to submit a revised version of the manuscript that addresses the points raised during the review process.

We look forward to receiving your revised manuscript.

Kind regards,

Novel N Chegou, Ph.D

Academic Editor

PLOS ONE

Journal Requirements:

Additional Editor Comments (if provided): Major Revision

Reviewers' comments:

Reviewer's Responses to Questions

**Comments to the Author**

1. Is the manuscript technically sound, and do the data support the conclusions?

Reviewer #1: Partly

Reviewer #2: No

Reviewer #3: No

2. Has the statistical analysis been performed appropriately and rigorously? 

Reviewer #1: I Don't Know

Reviewer #2: No

Reviewer #3: No

3. Have the authors made all data underlying the findings in their manuscript fully available?

Reviewer #1: Yes

Reviewer #2: No

Reviewer #3: Yes

4. Is the manuscript presented in an intelligible fashion and written in standard English?

Reviewer #1: Yes

Reviewer #2: No

Reviewer #3: Yes

5. Review Comments to the Author

Reviewer #1: In the manuscript by Dr. Tjernberg et al, the authors evaluated if detemining of selected immune response-related human protein biomarkers in serum using a proximity extension assay could be used to differentiate between patients with active Lyme borreliosis and healthy controls. The study seaks answers to an important clinical question i.e. how one could exclude with high probabilty the possibility of an active borrelial infection (and defer antibiotics) in patients with clinicaly suspected Lyme borreliosis, but not fulfiling criteria for definite LB. A new diagnostic approach beside serology would be of great value particularly in LB endemic areas with high background Borrelia seropositivity.

I have listed some concerns regarding the study design in the attached manuscript. Furthermore, I think the rational/reasons for selecting the particular set of immune response-related human protein biomarkers should be provided in the Introduction part. The comparative group of healthy blood donors could be described in more detail or defined more specificaly e.g. did seropositive asymptomatic controls had documented LB in the past? As I understand, the approach described would be a valuable addition to Borrelia serologic tests therfore, I would propose to make also a comparison between patients with active LB and those who are Borrelia seropositive, but do not have LB.

Reviewer #2: The manuscript by Tjernberg et al. addressed an important problem, which is that of discrimination between active Lyme borreliosis from resolved Borrelia infection after exposure. The paper however is poorly writing and needs major revisions before being considered for publication. There are major concerns associated with the study design and the proper discrimination of false positive.

Major concerns

Study design

There is logical and/or methodological problem with the study design, as the category ‘previous exposure’ can be included under ‘ongoing LB infection’. This is; patients with ‘previous exposure’ could have an ‘ongoing LB infection’. How the authors rule out that ‘blood donors with’ ‘anti-Borrelia antibodies’ did not had an ‘ongoing LB infection’? Are the authors referring to ‘acute’ vs. ‘chronic’ LB? In either case, the patients included in the study do not seem to be properly classified and/or the classification does not separate individuals beyond ‘seropositive’ or ‘seronegative’ to LB. Therefore, the question set by the authors need revision.

Methodology

It is known that several Borrelia burgdorferi proteins cross-react with that from other bacteria (see e.g., Eur J Clin Microbiol Infect Dis. 1992;11(3):224-32). Particularly, excessively high rates of false positive results of ELISA tests for Lyme Borreliosis (> 90% of positives were found to be false positives) were significantly reduced after adsorption of patient sera with Escherichia coli antigens (see e.g., J Rheumatol. 1995;22(4):684-8). Not clear how the authors dealt here with ‘false positive’ samples in the present study. For example, will sera with anti-Borrelia antibodies be still positive if adsorption with Escherichia coli antigens is performed?

In general, the methodology is poorly described. Scientific papers must include details so that others can repeat the experiments and/or analysis published.

Minor comments

Abstract

Line 58: It is confusing that the authors set to ‘discriminate an ongoing Lyme borreliosis (LB) infection from a previous exposure.’ (Lines 53-54), however, the cohort of LB patients (as mentioned in line 58) does not include patients with ‘ongoing LB infection’ and other with ‘previous exposure’. If these two groups of patients were included, this has to be mentioned here.

Line 65: Not clear what ‘immune-related proteins’ mean.

Line 68: These abbreviations are not very helpful, even less in an abstract.

Methods

Line 112: Based on what criteria was the ‘subset of serum samples from a previous prospective study’ selected? What the authors mean by ‘well characterised adult LB’? Provide further details.

Lines 116-117: Please, include general information on the location of origin of the ‘patients’ and ‘blood donors’.

Lines 118-120: ‘Classification and confirmation of LB diagnosis’ is poorly described. Improve.

Lines 120-122: Provide details about this kit. What antigen(s) the kit uses? What dilution of reagents was used?

Line 125: What ‘symptoms or signs of disease’ were considered?

Lines 130-141: This section has to be separated into different subsections and each one of these detail the methodologies used. For example, what ‘multiple analytes are detected in parallel’? How the ‘unique strands of DNA oligonucleotides’ and ‘individually paired matched antibodies’ were generated? How were these specific to LB? How the Borrelia proteins tested in the assay were obtained? How the ‘quantitative PCR’ was performed? What adaptors were used?

Reviewer #3: 1. According to the objectives, the authors aimed to investigate immunological protein biomarkers to search for specific patterns for active Lyme borreliosis. They conclude that levels of a number of them differed significantly from healthy blood donors, whether or not previously exposed to Borrelia.

My major comment is that it is not surprising that patients with an active infectious disease differ from healthy controls with regard to some immunological protein biomarkers. This is also the case for CXCL13 in CSF for active neuroborreliosis. However, in that entity patients with neuroborreliosis have also extensively compared with patients with other infectious and non-infectious diseases and high levels of this specific biomarker are rather specific for neuroborreliosis.

In the present paper, any comparison of the relevance of these immunological protein biomarkers in Lyme borreliosis and in other infectious and non-infectious diseases is lacking. Whereas I agree that in a pilot study like this one it may not yet be relevant to test such patient groups, I suppose that these biomarkers have also been studied in other diseases. Therefore I think it is necessary to include a section about what is known of the supposedly relevant biomarkers in other diseases. Are they non-specific and increased with inflammation, like CRP, or are their levels increased –or decreased- especially in Lyme borreliosis?

2. Some of the blood donors were seropositive for LB, others were not. This is extensively mentioned in the abstract. However, it is not at all stated whether seropositive donors differed from seronegative donors with regard to any marker. Probably, this is not the case, but the authors should clarify this. If any difference would exist between these populations, follow-up procedures comparing all blood donors and LB patients would not be valid.

3. The title suggests that the paper was aimed at comparing active Lyme borreliosis with previous Borrelia exposure. However, as stated in 2., patients with active LB are not compared to patients with previous Borrelia exposure – and follow-up samples of treated LB patients would be more relevant to provide insight in differences between active and past LB. The title should rather be “A novel laboratory approach to discriminate active Lyme borreliosis form healthy controls” or more appropriate, “Identification of potential biomarkers in active Lyme borreliosis”

4. Another limitation of the study is that most patients (41/52) were patients with neuroborreliosis. The number of patients with arthritis, acrodermatitis and erythema migrans was small and results might not be applicable to those patients. The authors might also consider to calculate the AUC for the comparison blood donors – neuroborreliosis patients, since this is a more clearly described clinical entity and this comparison could be more relevant for clinical practice.

5. It is not clear what is represented in table S1, probably same as in table 2?

Minor comment:

L.172: sentence is incomplete.

6. PLOS authors have the option to publish the peer review history of their article (what does this mean?). If published, this will include your full peer review and any attached files.

Reviewer #1: **Yes: **Daša Stupica

Reviewer #2: No

Reviewer #3: **Yes: **Alje P. van Dam

---

## [Author Response · Author response to Decision Letter 0]

30 May 2023

We thank the reviewers for valuable comments and suggestions in order to improve the quality of the manuscript and refer to the document responses to reviewers for point-by-point responses. 

Best regards, 

Ivar Tjernberg, MD PhD Associate professor

---

## [Decision Letter · Decision Letter 1]

8 Jun 2023

Identification of potential biomarkers in active Lyme borreliosis

PONE-D-23-05893R1

Dear Dr. Tjernberg

We’re pleased to inform you that your manuscript has been judged scientifically suitable for publication and will be formally accepted for publication once it meets all outstanding technical requirements.

Kind regards,

Novel N. Chegou, Ph.D

Academic Editor

PLOS ONE

Additional Editor Comments (optional):

Reviewers' comments:

Reviewer's Responses to Questions

**Comments to the Author**

1. If the authors have adequately addressed your comments raised in a previous round of review and you feel that this manuscript is now acceptable for publication, you may indicate that here to bypass the “Comments to the Author” section, enter your conflict of interest statement in the “Confidential to Editor” section, and submit your "Accept" recommendation.

Reviewer #1: All comments have been addressed

Reviewer #2: All comments have been addressed

2. Is the manuscript technically sound, and do the data support the conclusions?

Reviewer #1: Yes

Reviewer #2: Yes

3. Has the statistical analysis been performed appropriately and rigorously? 

Reviewer #1: I Don't Know

Reviewer #2: Yes

4. Have the authors made all data underlying the findings in their manuscript fully available?

Reviewer #1: Yes

Reviewer #2: Yes

5. Is the manuscript presented in an intelligible fashion and written in standard English?

Reviewer #1: Yes

Reviewer #2: Yes

6. Review Comments to the Author

Reviewer #1: (No Response)

Reviewer #2: Thank you to the authors for addressing the previous concerns positively............................

7. PLOS authors have the option to publish the peer review history of their article (what does this mean?). If published, this will include your full peer review and any attached files.

Reviewer #1: No

Reviewer #2: No

---

## [Editor Report · Acceptance letter]

19 Jun 2023

PONE-D-23-05893R1 

Identification of potential biomarkers in active Lyme borreliosis 

Dear Dr. Tjernberg:

I'm pleased to inform you that your manuscript has been deemed suitable for publication in PLOS ONE. Congratulations! Your manuscript is now with our production department. 

Kind regards, 

on behalf of

Prof Novel Njweipi Chegou 

Academic Editor

PLOS ONE